

# Patterns of change in α and β taxonomic and phylogenetic diversity in the secondary succession of semi-natural grasslands in the Northern Apennines

Lorenzo Lazzaro[1], Lorenzo Lastrucci[2], Daniele Viciani[1], Renato Benesperi[1], Vincenzo Gonnelli[3] and Andrea Coppi[1]

[1] Department of Biology, University of Florence, Florence, Italy
[2] Natural History Museum, Botany, University of Florence, Florence, Italy
[3] Unaffiliated, Pieve Santo Stefano, Italy

## ABSTRACT

We studied the secondary succession in semi-natural grasslands (dry grasslands and hay meadows) located in the eastern side of the Tuscan Apennines (Tuscany, Central Italy). We compared these habitats, investigating: (i) the changes in species richness, composition and phylogenetic diversity during the succession; (ii) whether the trends in species loss and species turnover in taxonomic diversity matched those in phylogenetic diversity. We performed a stratified random sampling, in a full factorial design between habitat type and succession stage (60 sampled plots, 10 × 2 types of habitat × 3 stages of succession). We constructed a phylogenetic tree of the plant communities and compared the differences in taxonomic/phylogenetic α- and β-diversity between these two habitats and during their succession. We identified indicator species for each succession stage and habitat. Looking at α-diversity, both habitats displayed a decrease in species richness, with a random process of species selection in the earlier succession stages from the species regional pool. Nevertheless, in the latter stage of dry grasslands we recorded a shift towards phylogenetic overdispersion at the higher-level groups in the phylogenetic tree. In both habitats, while the richness decreased with succession stage, most species were replaced during the succession. However, the hay meadows were characterized by a higher rate of new species' ingression whereas the dry grasslands became dominated with *Juniperus communis*. Accordingly, the two habitats showed similar features in phylogenetic β-diversity. The main component was true phylogenetic turnover, due to replacement of unique lineages along the succession. Nevertheless, in dry grasslands this trend is slightly higher than expected considering the major importance of difference in species richness of dry grasslands sites and this is due to the presence of a phylogenetically very distant species (*J. communis*).

## INTRODUCTION

Secondary semi-natural grasslands represent important components of European cultural landscapes. They derive from centuries of traditional land use, mainly linked to grazing

Corresponding author
Lorenzo Lazzaro,
lorenzo.lazzaro@unifi.it

by livestock (pastures) or hay-making (meadows) (*Dengler et al., 2014*; *Janišová et al., 2011*). Many secondary grassland vegetation types are considered habitats worthy of conservation and are listed in European and national protection directives and laws, such as the Low altitude hay meadows and the *Festuco-Brometea* dry grasslands (respectively codes 6510 and 6210, according to European Council Directive 92/43/EEC). In Europe, because of recent cultural changes, secondary grasslands and meadows have displayed an overall tendency to evolve into shrublands and woodlands through natural secondary successions (*Dengler et al., 2014*; *Monteiro et al., 2011*; *Peco, Sánchez & Azcárate, 2006*; *Peco et al., 2012*). Perennial grasslands undergo a vegetation dynamism leading to the gradual transition from herbaceous coenoses belonging to syntaxonomical classes such as *Festuco-Brometea* or *Molinio-Arrhenatheretea* (*Allegrezza & Biondi, 2011*; *Biondi et al., 1995*) to shrub coenoses belonging, especially in central Italy, mainly to the *Rhamno-Prunetea* class (*Biondi, Allegrezza & Guitian, 1988*). The shrubland formations are sometimes preceded by intermediate phases dominated by other herbaceous species such as *Brachypodium rupestre* (*Assini et al., 2014*) and mostly originate from the species that form the fringes of woods surrounding grasslands (*Biondi, Allegrezza & Guitian, 1988*; *Poldini, Vidali & Zanatta, 2002*).

Such succession, with the consequent loss in species, has been widely studied from a landscape viewpoint (*Bracchetti, Carotenuto & Catorci, 2012*; *Rocchini et al., 2006*; *Viciani et al., 2018*). Many authors have also focused on the functionality of the communities, elucidating the central role of competition in the loss of species characterizing the early phases of succession (*Csergő, Demeter & Turkington, 2013*; *Lepš, 1999*; *Peco et al., 2012*). Nevertheless, less is known regarding the changes occurring in the communities with respect to trends in β-diversity phylogenetic relationships, also concerning their link with changes at the taxonomic level. Indeed, according to *Chase & Myers (2011)*, β-diversity can provide considerable insights into the importance of deterministic and stochastic processes in generating community structure along spatial and ecological gradients. In addition, the use of molecular phylogenies may be helpful in analyzing the forces that influence patterns of biodiversity and biogeography and in depicting the interactions among co-occurring species (*Selvi, Carrari & Coppi, 2016*). Indeed, in the last decades, the use of molecular phylogeny has increased widely for ecological purposes, contributing also to the emerging area of community phylogenetics (*Webb et al., 2002*). One of the multiple ways to use the phylogenetic information consists in the measure of the phylogenetic overdispersion or clustering of the community in relation to the variation of the habitat conditions (*Erickson et al., 2014*; *Qian, Hao & Zhang, 2014*; *Selvi, Carrari & Coppi, 2016*). Recently, many authors (*Webb, 2000*; *Webb et al., 2002*; *Kembel, 2009*) have highlighted that the observed patterns of phylogenetic structure of the communities could be used to understand the processes of community assembly, particularly linking patterns of phylogenetic clustering and overdispersion with the processes of habitat filtering and competitive exclusion. Indeed, the use of phylogenetic patterns as proxies for the processes of community assembly is rapidly raising concerns linked to the assumptions underlying such approach, often only weakly supported. In particular, these assumptions regard the existence of an actual correlation between

measures of phylogenetic dispersion and trait dispersion, the idea that trait similarity would enhance competition and that competition necessarily causes species exclusion from the community and that community assemblages are in status of equilibrium (*Gerhold et al., 2015*; *Prinzing, 2016*). Furthermore, *Kraft et al. (2015)* reported a misuse of the concept of environmental filtering, considering that most empirical studies hardly distinguish the effects of abiotic factors from those of biotic interactions and often overestimate the role of the environment in shaping communities.

In our study area, the Tuscan Apennines, the co-occurrence of species in secondary grasslands is driven by different agricultural management in different geo-morphological conditions. The areas with low slope inclination and relatively fertile and deep soils are at first subjected to machining and sowing, mainly with plants that increase the nutrient value in the soil (i.e., *Medicago sativa, Onobrychis viciifolia*). Then, herbaceous natural grassland species (many from the families *Poaceae, Fabaceae* and *Asteraceae*) start to colonize these communities, which in a few years become semi-natural hay meadows (*Ubaldi, 2003*). On the other hand, areas on steeper slopes with shallow (sometime even rocky) soils are used as pastures and become dry grasslands. Also in our area of study, the main characteristic differentiating the two habitat types is the stand geomorphology, with significantly steep dry grasslands and almost flat hay meadows: these differences may affect the water capacity, structure and fertility of soils.

We expect that these ecological differences may also affect the secondary succession, leading to different species assemblages. Indeed, as anthropic pressures are relieved, the succession of these habitats is generally characterized by intermediate phases dominated by herbaceous species, leading finally to quite different shrub formations. In particular, while shrubland following hay meadows are dominated by several species of broadleaf shrubs, dry grasslands are dominated by *Juniperus communis*, a distantly related stress-tolerant species (*Pierce et al., 2017*).

The study of taxonomic, phylogenetic and functional plant (α and β) diversity along secondary succession has already been the object of studies (*Purschke et al., 2013*), which highlighted that the relative importance of assembly processes had changed over time, but with contrasting patterns of temporal change in the different facets of diversity. Nevertheless, while *Purschke et al. (2013)* observed a general increase in taxonomic, phylogenetic and functional alpha-diversity during succession, we expect a decrease in species taxonomic diversity, not necessarily linked to a decrease in phylogenetic diversity. Moreover, *Purschke et al. (2013)* reported a predominant role of abiotic filtering in community assembly during the early stages of grassland succession, whereas the relative importance of competitive exclusion appears to have increased towards the later succession stage. Conversely, a major role of competition by dominant tall grasses as already been described for the early stages of succession in Apennine grasslands communities (*Corazza et al., 2016*). According to these differences, we sought to (i) assess the relative changes in α- and β-diversity taxonomic and phylogenetic diversity, also identifying the indicator species in the different stages of the succession and (ii) assess whether the trends in species loss and species turnover in taxonomic diversity matches those in phylogenetic diversity. Moreover, we further hypothesize that *Juniperus communis*

may have a pivotal role, driving a certain degree of overdispersion of the phylogenetic structure of the communities in dry grasslands, considering its distant relatedness.

Toward these aims, we conducted a sampling of dry grasslands and hay meadows plant communities in the Tuscan Apennines, adopting a chrono-sequential approach to reconstruct their dynamic changes, assessing the changes in of α- and β-diversity within and between three succession stages of these two habitats.

## MATERIALS AND METHODS

### Study area

The study area is located in the Adriatic side of the south-eastern Tuscan Apennines (Tuscany, Central Italy, 43.691838°N 12.111936°E). The area is managed by the Raggruppamento Carabinieri Biodiversità - Reparto Carabinieri Biodiversità Pieve Santo Stefano. The most represented geological units are the clays of the Sillano Formation, deposits of landslide debris and to a lesser extent, the sandstones of Falterona Mount (*Carmignani et al., 2013*). Climate belongs to the Temperate Oceanic Bioclimate (*Pesaresi, Biondi & Casavecchia, 2017*).

### Sampling design

The study was carried out exploiting a random/stratified sampling design involving two layers: type of habitat (hereafter Habitat) and stage of succession (hereafter Succession). To identify and map these layers correctly, we performed an analysis of orthophotos of the study area using the QGis software (version 2.14.21, QGIS.ORG project). We used the cover percentage of scrub to distinguish: (i) dry grasslands and hay meadows (shrub presence not detectable), (ii) mixed typologies (dry grasslands/hay meadows with shrub cover <50%) and (iii) shrubland (shrub cover >50%). Using also the historical knowledge concerning the landscape management provided by the local administrator, we defined the sampling layers as follow: (1) Habitat (corresponding to the land use) distinguishing between dry grasslands and hay meadows and (2) Succession, distinguishing between (i) active (hereafter Managed) (ii) short-time abandoned (hereafter Transition) and (iii) long-time abandoned areas (hereafter Abandoned). The stage of succession is used as a proxy for the time from abandonment of the typical land management practices, considering that an increase of the cover of shrubs corresponds to the increase of time following abandonment. Hence, we performed a random selection of 10 square plots of $2 \times 2$ m for each stratum in a full factorial design, leading to 60 sampled plots ($10 \times 2$ types of habitat $\times$ 3 stages of succession). In each plot, we performed a floristic sampling, recording the presence/absence of vascular plants. Voucher specimens were collected for identification in the laboratory and further samples were collected for the genetic analyses.

### Selection of molecular markers, DNA isolation, sequence alignment and tree reconstruction

Phylogenetic diversity of the spermatophyte communities was inferred from the analysis of three markers of the nuclear ribosomal DNA, the ITS1-5.8S and ITS2 regions. These markers, widely used for phylogenetic studies both in plants and fungi, have shown a great

discriminatory power at low taxonomic levels (*Feliner & Rosselló, 2007*; *Hollingsworth, Graham & Little, 2011*), supporting this region as a core barcode for spermatophytes (*Li et al., 2011*) and hence potentially usable as a proxy for evolutionary relationships. We assembled a sequence dataset by retrieving accessions of the sampled species from GenBank (http://www.ncbi.nlm.nih.gov/) to construct a tree resolved at the species level. Molecular analyses were performed for 13 species for which no accessions were available in the GenBank. Isolation of genomic DNA followed a modified 2 × CTAB protocol successfully adopted in previous studies using molecular tools (*Coppi, Mengoni & Selvi, 2008* and Appendix S1 for further details).

The final dataset of the sampled species consisted of 147 accessions (Table S2). Taxa from Gnetales, Ginkgoales, Cycadales, Pinales and Cupressales were added in order to obtain a correct alignment for Gymnosperms, whereas four taxa from Polypodiales and Salviniales were added as outgroups (Table S1). Multiple alignment of the ITS-5.8S dataset was performed with MAFFT (v. 7.0, *Katoh & Standley, 2013*) adopting the parameterization typically used for nucleotide sequences (200PAM/$k = 2$, gap penalty = 1.53; offset = 0.0), considering that we aligned very distant species. We followed a step by step multi-alignment procedure: (1) taxa were grouped at the order level and aligned using the Q-INS-i strategy, checking each multi-alignment by visual inspection with BioEdit; (2) the multi-alignments were merged at higher ranks using the *merge* option in MAFFT, obtaining separate multi-alignments for Eudicots, Monocots, Gymnosperms and for the outgroup; (3) these four multi-alignments were finally merged again. The alignment was used to build a phylogenetic tree with a maximum likelihood (ML) approach by means of RA×ML (*Stamatakis, 2006*) via the CIPRES supercomputer cluster (http://www.phylo.org/), using 1,000 maximum searches. The topology of phylogenetic inference was constrained at the family level using as backbone the tree *slik2015* (*Slik et al., 2018*) available in Phylomatic vers. 3 (http://phylodiversity.net/phylomatic/). This topology is mainly based on the APG III phylogeny, further resolved up to genus level using the species-level phylogeny in *Zanne et al. (2014)*, placing at the base of their respective families genera not present in *Zanne et al. (2014)*, see *Slik et al. (2018)*. The resulting topology is highly consistent with the hypothesis in *Magallón et al. (2015)*, which was subsequently used to date the phylogeny, except for the clade involving the *Malpighiales* and the one involving *Boraginales*, *Solanales* and *Gentianales*. For these clades we followed the tree *slik2015*. The statistical support to the nodes was estimated using the bootstrap method (1,000 iterations). Finally, to obtain an ultrametric tree, we calibrated our phylogeny dating the node ages according to *Magallón et al. (2015)*, adopting a Molecular Dating approach throughout Penalised Likelihood estimation via the *chronos* function of ape v5.1 R package (*Paradis, Claude & Strimmer, 2004*).

## Evaluation of trends in species richness and phylogenetic relatedness (α-diversity)

We evaluated the changes in taxonomic species richness (SR) and in three indices allowing the assessment of different features of phylogenetic α-diversity. We used the Phylogenetic Diversity (PD) as a measure of the amount of phylogenetic richness in the

communities (how much) and the Net Relatedness Index (NRI) and the Nearest Taxon Index (NTI) to provide information regarding the phylogenetic divergence within the communities (how different they are) (*Tucker et al., 2017*).

Faith's PD (*Faith, 1992*) represents the simplest measure of the cumulative evolutionary age in a community, but it is highly correlated to species richness. Thus, we adopted its Standardized Effect Size index that is generally considered unaffected by species richness (*Pavoine et al., 2013*; *Swenson, 2014*; but see *Sandel, 2018*) and indicates whether the observed PD is different from what would be expected by chance. To allow an interpretation of results comparable with NRI and NTI, we multiplied PD.ses by −1. We defined NTI and NRI as Standardized Effect Size indices of Mean Nearest Taxon Distance (MNTD) for taxa in a community and Mean Pairwise Distance separating taxa in a community (MPD). MNTD is calculated as the mean of the smallest non-diagonal value in the pairwise distance matrix for each species and is a measure of the branch-tip phylogenetic clustering of the species in the community (*Webb et al., 2002*). It describes the phylogenetic relatedness among species, focusing on the distal part of the tree, thus involving lower taxonomical levels. MPD estimates the average phylogenetic relatedness between all possible pairs of taxa in an assemblage, calculated as the mean of the non-diagonal elements in the pairwise distance matrix (*Webb, 2000*). It is a measure of the relationship at the higher-level groups in the phylogenetic tree. NTI and NRI indicate whether the values of MPD and MNTD differ from what would be expected by chance. Positive values of NRI and NTI indicate that observed phylogenetic distances are lower than expected and that phylogenetic clustering of species occurs. Conversely, negative values of such indices indicate phylogenetic over-dispersion or evenness. In general terms, NTI and NRI values higher/lower than 1.96/−1.96 are usually considered indicators of significant patterns. All standardized indices (PD.ses, NRI and NTI). All standardized effect sized indices (PD.ses, NRI and NTI) were calculated using a comparison with *fixed-fixed* null models, which maintain both species richness and species abundance across sites and tend to exhibit low type I and II error rates (*Miller, Farine & Trisos, 2017*). The null model matrices were randomized using the "independent-swap" algorithm by *Gotelli (2000)*, which is well suited for presence/absence community matrices.

We studied the variation SR and PD.ses, NTI and NRI in a two-way ANOVA design considering Habitat and Succession as explanatory variables, also taking into account their interaction effect. We further explored the differences in the levels of the significant terms performing a Post-Hoc Tukey Test.

To assess the role of *Junipers communis* in the phylogenetic α-diversity patterns found, we repeated the analyses on PD.ses, NTI and NRI excluding this species and keeping the same design described above.

## Trends of compositional shifts (β-diversity)

To assess the variations in species composition of plots, we ran a comparison among a detrended correspondence analysis (DCA) and a canonical correspondences analysis (CCA) on the plot species composition. Thus, we checked the efficiency of constrained axis

to catch the variation explained by unconstrained axis, as provided in the software Canoco 5 (*Ter Braak & Šmilauer, 2012*) and following *Šmilauer & Lepš (2014)*. In DCA the axes were detrended by segment, adopting default options in Canoco 5 In the CCA, Habitat and Succession were used as explanatory variables and the significance of the constrained axes was tested with 4,999 unrestricted permutations.

We used the methodological framework developed by *Podani & Schmera (2011)* to evaluate the trends in β-diversity components during the succession. This methodology allows the partition of pairwise gamma diversity into three complementary indices, measuring Similarity, relative Richness Difference and relative Species Replacement (respectively S, D and R) and accordingly is referred to as SDRSimplex approach (see also Appendix S2 for further description). Pairwise-comparisons regarded plots of the same habitat, spanning along the succession. The SDRSimplex results were projected in a ternary plot. Finally, we used a Nonparametric Kruskal–Wallis test to check the significance of the differences among habitats.

To assess the role of particular species in the species turnover, we carried out an Indicator Species Analysis (ISA, *Dufrêne & Legendre, 1997*). The ISA allows computing an indicator value $d$ (ranging between 0 and 100) of each species as the product of the relative frequency and relative average abundance of species in clusters. The analysis also produces a significance value, representing the probability of obtaining a $d$ value as high as that observed over 1,000 iterations. We conducted the analyses considering each stage of succession of the two habitats as a separate cluster.

We used a three dimension Non-Metric Multidimensional Scaling (NMDS) ordination based on the UniFrac index distance matrix, to assess how different were the communities from the phylogenetic point of view. UniFrac is a phylogenetic diversity-based dissimilarity index that measures the proportion of evolutionary history unique to each community and is calculated as the total branch length unique to each community relative to the total branch length linking all species in both communities (*Lozupone & Knight, 2005*).

Furthermore, we studied the evolutionary dissimilarity between communities along the succession (phylogenetic β-diversity), adopting the approach described in *Leprieur et al. (2012)* as an improvement of PhyloSor index. This index expresses to what extent the compared communities are composed by related species rather than by species that share no branch in the phylogeny and can be separated in two components accounting for "true" phylogenetic turnover (PhyloSor$_{Turn}$) and phylogenetic diversity gradients (PhyloSor$_{PD}$). In addition, we also analyzed the standardized effect size of such indices (i.e., SES.PhyloSor, SES.PhyloSor$_{Turn}$ and SES.PhyloSor$_{PD}$), which describe whether two communities are phylogenetically more or less dissimilar than what is expected given their taxa dissimilarity. These indices are obtained via comparison with a null model in which species are randomized across the tips from the tree while holding constant species richness and compositional beta diversity in 999 simulations. Again, we used a Nonparametric Kruskal–Wallis test to check the significance of the differences among the two habitat types.

**Table 1 Analysis of variance table for the effect of Habitat and Succession on indices of taxonomic and phylogenetic α-diversity.** Species Richness (SR), Standardized Effect Size of Faith's Phylogenetic Diversity (PD.ses), Net Relatedness Index (NRI) and Nearest Taxon Index (NTI).

| Response | Factors | Df | ResDf | Sum Sq | *F* value | *P* value |
|---|---|---|---|---|---|---|
| SR | Habitat | 1 | 54 | 22.82 | 2.067 | 0.156 |
| | Succession | 2 | 54 | 1,930.23 | 87.458 | <0.001*** |
| | Habitat: Succession | 2 | 54 | 54.03 | 2.448 | 0.096 |
| PD.ses | Habitat | 1 | 54 | 9.49 | 8.34 | 0.006** |
| | Succession | 2 | 54 | 55.60 | 24.43 | <0.001*** |
| | Habitat: Succession | 2 | 54 | 1.87 | 0.83 | 0.440 |
| NRI | Habitat | 1 | 54 | 21.33 | 17.77 | <0.001*** |
| | Succession | 2 | 54 | 50.27 | 21.33 | <0.001*** |
| | Habitat: Succession | 2 | 54 | 17.08 | 7.11 | 0.002** |
| NTI | Habitat | 1 | 54 | 3.16 | 2.92 | 0.093 |
| | Succession | 2 | 54 | 48.82 | 21.19 | <0.001*** |
| | Habitat: Succession | 2 | 54 | 1.29 | 0.64 | 0.554 |

**Notes:**
** Significance codes: *P* value < 0.01.
*** Significance codes: *P* value < 0.001.

Furthermore, to assess the role of *Junipers communis* in the phylogenetic β-diversity patterns found, we repeated the analyses excluding this species and keeping the same design described above.

All ordination analyses (DCA, CCA and NMDS) and relative graphs were made using the software Canoco 5 vers. 5.12 (*Ter Braak & Šmilauer, 2012*). All other analyses were made using R software (version 3.5, The R Foundation for Statistical Computing, Vienna, Austria, www.R-project.org) and relative graphs were produced with ggplot2 vers. 2.2.1 (*Wickham, 2009*). The ISA was conducted using the package labdsv (R package version 1.8-0, https://CRAN.R-project.org/package=labdsv). Phylogenetic β-diversity indices were calculated exploiting the R functions developed by *Leprieur et al. (2012)*. Phylogenetic α-diversity metrics were obtained with the package metricTester vers. 1.3.6 (*Miller, Farine & Trisos, 2017*).

# RESULTS

The sampling resulted in 147 species (Table S2), with 69 shared species among the two habitats and 39 species exclusive to each habitat (tot. 108 species in both habitats). Species richness varied from 5 to 29 species per plot, in Abandoned dry grasslands and Managed dry grasslands, respectively. The resulting phylogenetic tree is shown in Fig. S1 (see Appendix S3 for the tree in Newick format).

Species richness was comparable between the two habitats and decreased in both during the succession (Table 1; Fig. 1A). Conversely, PD.ses differed significantly among the habitats, with lower values in dry grasslands and during the succession, with a decrease in the latter stage (Table 1; Fig. 1B). Regarding NRI, the two habitats showed different trends during the succession (significant interaction Succession × Habitat, Table 1).
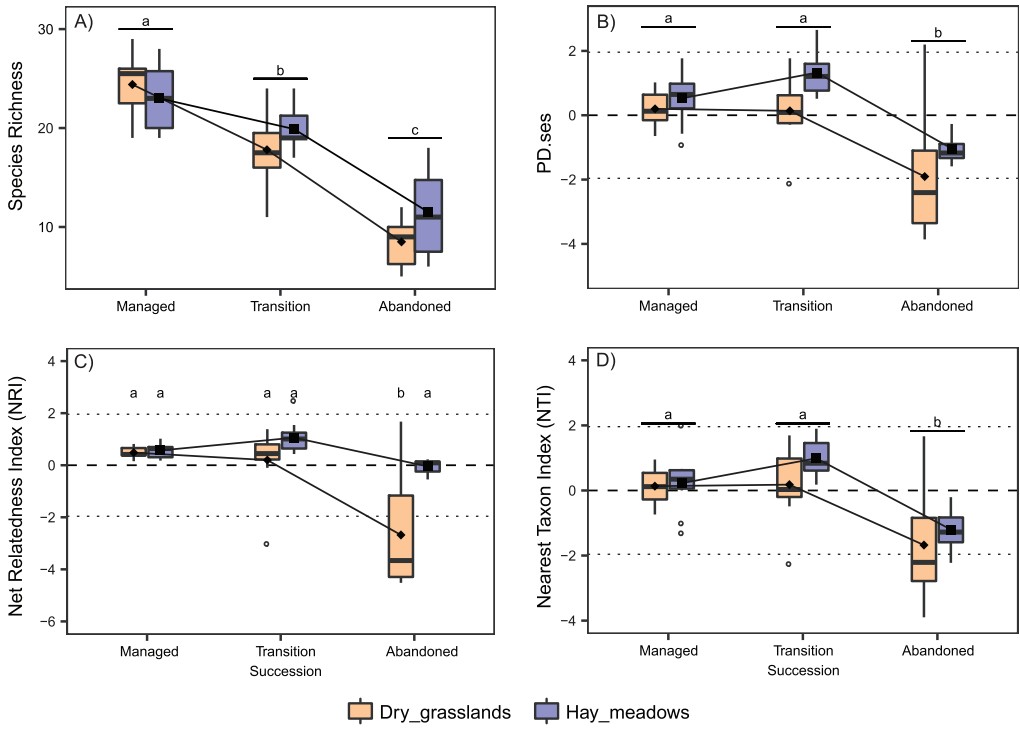

**Figure 1 Interaction plots for the variation in taxonomic and phylogenetic α-diversity of the 60 sampled plots according to Habitat and Succession.** (A) Species Richness (SR). (B) Standardized Effect Size of Faith's Phylogenetic Diversity (PD.ses). (C) Net Relatedness Index (NRI). (D) Nearest Taxon Index (NTI). Different letters indicate significant differences ($p < 0.05$) after a Post Hoc Tukey's test conducted in (A), (B) and (D) between the levels of the factor Succession and in (C) between the levels of the interaction Succession × Habitat, according to the ANOVA results.

Indeed, while they displayed comparable values at earlier succession stages with values indicating random processes of species selection, dry grasslands shifted in the latter stage to a significant overdispersion of plant composition, with the mean NRI value below the critical threshold of −1.96 and significantly different from the one of hay meadows (Fig. 1C). On the other hand, the two habitats displayed a comparable NTI trend during the succession, with values in hay meadows generally higher (Succession and Habitat both significant, but no significant interaction, Table 1). Notwithstanding a significant drop in the latter stages of succession, in all three stages the mean NTI values remained between ±1.96, again indicating random processes of species selection (Fig. 1D).

The analyses concerning phylogenetic α-diversity conduced excluding *J. communis* showed quite a different scenario. PD.ses varied significantly between the two habitats and showed different trends during the succession (Interaction term Succession × Habitat significant, see Table 2), while both NTI and NRI varied significantly only along the succession with comparable trends between the two habitats (Table 2). Indeed, we recorded a steep increase in PD.ses in abandoned dry grasslands compared to the values obtained considering *J. communis*. As a result, PD.ses in dry grassland is more or less stable along the succession, while it varies significantly in hay meadows (Fig. 2A). On the

**Table 2 Analysis of variance table for the effect of Habitat and Succession on indices of phylogenetic α-diversity evaluated excluding from the analyses the species *Juniperus communis*.** Standardized Effect Size of Faith's Phylogenetic Diversity (PD.ses), Net Relatedness Index (NRI) and Nearest Taxon Index (NTI).

| Response | Factors | Df | ResDf | Sum Sq | *F* value | *P* value |
|----------|---------|-----|-------|--------|-----------|-----------|
| PD.ses | Habitat | 1 | 54 | 0.50 | 0.58 | 0.451 |
| | Succession | 2 | 54 | 28.29 | 16.30 | <0.001*** |
| | Habitat: Succession | 2 | 54 | 11.03 | 6.36 | 0.003** |
| NRI | Habitat | 1 | 54 | 0.39 | 2.05 | 0.158 |
| | Succession | 2 | 54 | 7.33 | 19.03 | <0.001*** |
| | Habitat: Succession | 2 | 54 | 1.00 | 2.61 | 0.083 |
| NTI | Habitat | 1 | 54 | 0.01 | 0.01 | 0.920 |
| | Succession | 2 | 54 | 25.69 | 17.09 | <0.001*** |
| | Habitat: Succession | 2 | 54 | 4.18 | 2.78 | 0.071 |

**Notes:**
** Significance codes: *P* value < 0.01.
*** Significance codes: *P* value < 0.001.

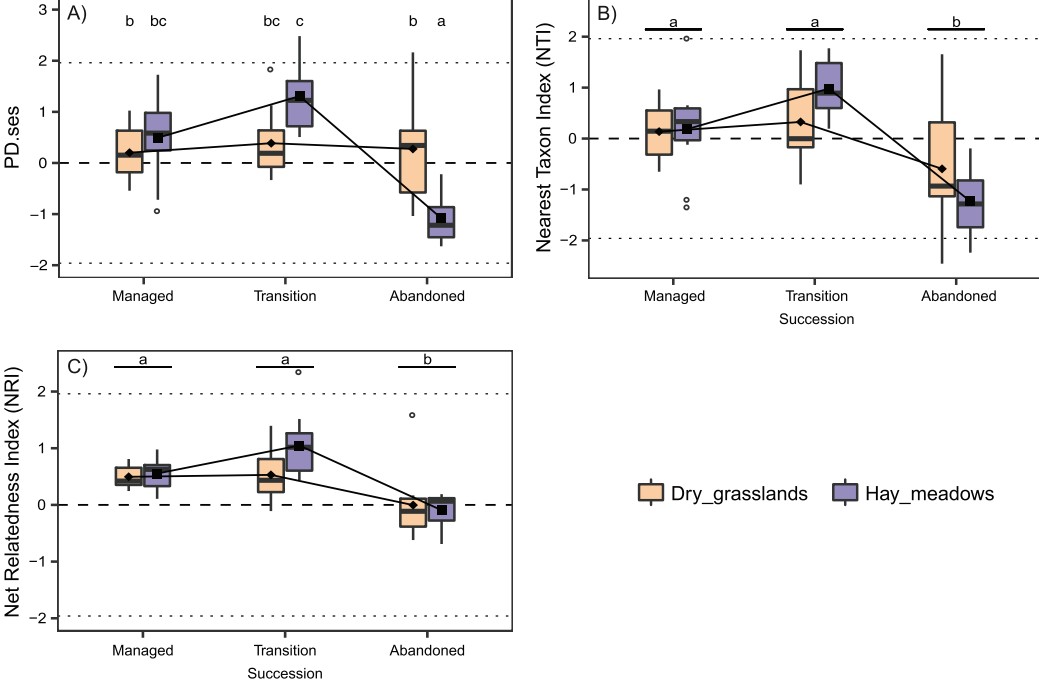

**Figure 2 Interaction plots for the variation in phylogenetic α-diversity of the 60 sampled plots according to Habitat and Succession evaluated excluding from the analyses the species *Juniperus communis*.** (A) Standardized Effect Size of Faith's Phylogenetic Diversity (PD.ses). (B) Net Relatedness Index (NRI). (C) Nearest Taxon Index (NTI). Different letters indicate significant differences (*p* < 0.05) after a Post Hoc Tukey's test conducted in (A) between the levels of the interaction Succession × Habitat and in (B) and (C) between the levels of the factor Succession, according to the ANOVA results.

contrary, phylogenetic overdispersion in dry grasslands clearly vanished, with both NTI and NRI higher than those *J. communis* and even closer to 0 than those of hay meadows, thus with a net predominance of random processes of species selection (Figs. 2C and 2D).

The species composition of two habitats resulted clearly separated along the succession. CCA constrained axes showed a very good efficiency, catching 85.5% and 100% of variation explained by DCA unconstrained axis. Total variation is 4.87, DCA first two axes explained 10.9% and 6.9% of it, while CCA ones 9.3% and 6.9% ($P < 0.001$). Plots resulted clearly differentiated in terms of species composition with a clear set of taxa specific to each habitat and each stage, with changes during the succession lying on the horizontal axis and differences among the two habitats on the vertical one (see Figs. 3A and 3B). According to the ordinations, the two habitats show common trends of species replacement, with new species coming in the transition stage and a further differentiation in the last stage.

The pairwise comparisons across the two habitats highlighted significant differences in the trends in species turnover among the succession. Indeed, they shared consistently low values of similarity (S) but were characterized by significantly different values of species replacement (R) and richness difference (D). In particular, dry grasslands displayed a higher D and a lower R than hay meadows (Fig. 4).

The ISA confirmed the presence of different trends in the numbers of species characterizing the succession stages and leading the succession. In the managed stage, the two habitats shared the same number of indicator species, but hay meadows presented a higher number of indicators species than dry grasslands in the following stages (17, 5 and 9 species in hay meadows vs. 17, 3 and 1 in dry grasslands in the Managed, Transitional and Abandoned stages, respectively, see Table S3).

The NMDS ordination on phylogenetic distance among plot allowed a good representation of the UniFrac distances (Stress criterion = 0.141). As for DCA and CCA, in the NMDS first axis reported the changes during the succession, while the habitat are separated along the second axis. Here, particularly the transitional stage of hay meadows show a high similarity with managed and transition stages of dry grasslands, while again the two habitats showed a relevant differentiation in the latter stage (Fig. 5).

Phylogenetic β-diversity was comparable among the two habitats and in both communities was mainly composed by phylogenetic turnover (Fig. 6A). Only PhyloSorPD was significantly different between the two habitats, being higher in dry grasslands than in hay meadows. None of the phylogenetic β-diversity standardized metrics differed significantly from what is expected given the taxa dissimilarity (all values between ±1.96, see Fig. 6C). Nevertheless, both SES.PhyloSor and the component SES.PhyloSorPD resulted significantly lower in dry grasslands than in hay meadows. The analyses conducted excluding *J. communis* from dry grasslands resulted in higher values of PhyloSor$_{PD}$ and lower ones of PhyloSorPD, both differing significantly between the two habitats (Fig. 6B). SES.PhyloSor$_{Turn}$ also showed significantly lower values in dry grasslands (Fig. 6D).

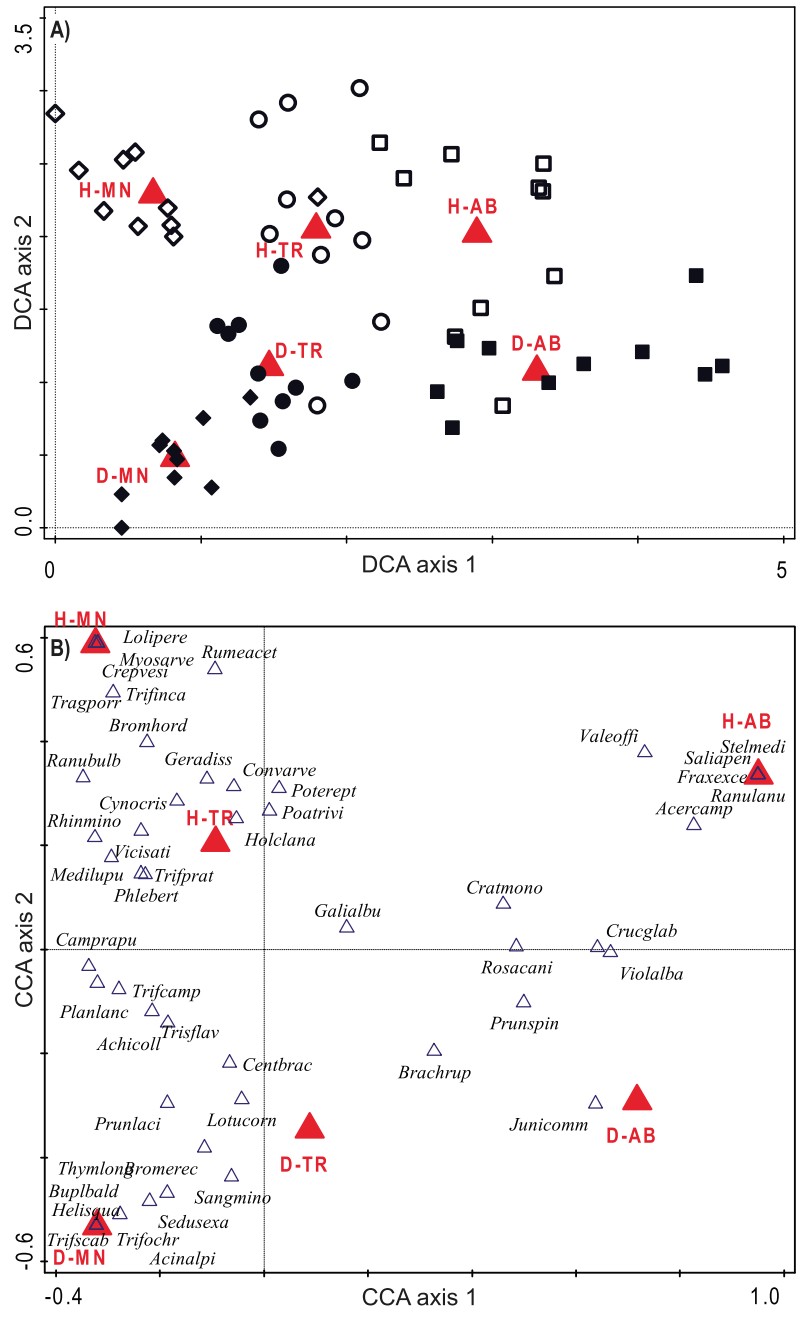

**Figure 3 Ordination diagrams considering species composition of plots.** (A) Plot distribution according to DCA considering their species composition. Empty symbols represent hay meadows and full symbols represent dry grasslands. Rhombus represent managed plots, circles transition plots and squares abandoned ones. (B) Plant species distribution obtained with CCA, only 50 best fitting species are shown; see also Table S3 for indicator species. Blue triangles represent the species. In both graphs, red triangles represent plot centroids according to Habitat type and Stage of Succession. H-MN, managed hay meadows; H-TR, transition hay meadows; H-AB, abandoned hay meadows; D-MN, managed dry grasslands; D-TR, transition dry grasslands; D-AB, abandoned dry grasslands.

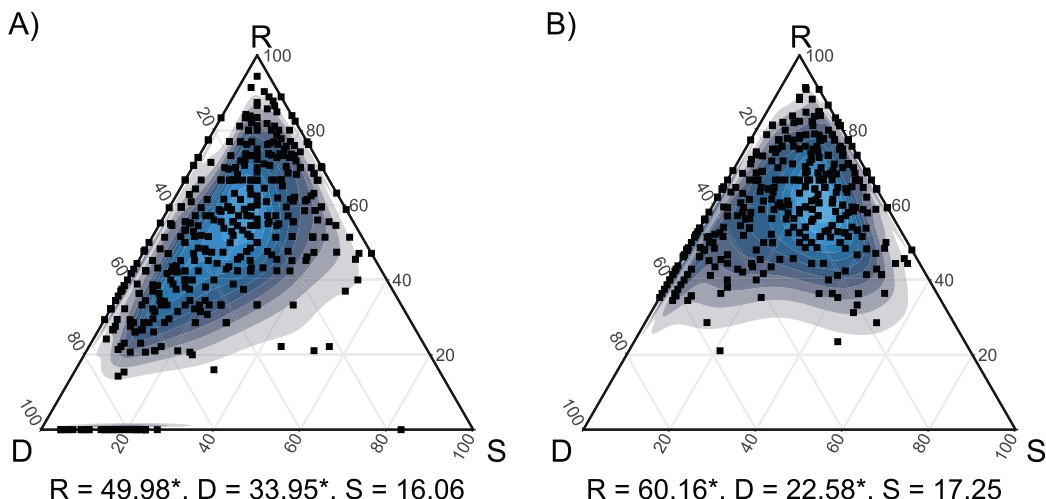

**Figure 4 SDR simplex ternary plots showing the variation in taxonomic β-diversity along the secondary succession of the two habitats.** (A) Dry grasslands. (B) Hay meadows. The abbreviations S, D and R refer to similarity, richness difference and species replacement, respectively. Mean values of S, D and R are reported. Values marked with * are significantly different at $P < 0.05$ according to a Kruskal–Wallis test performed between the two habitats.

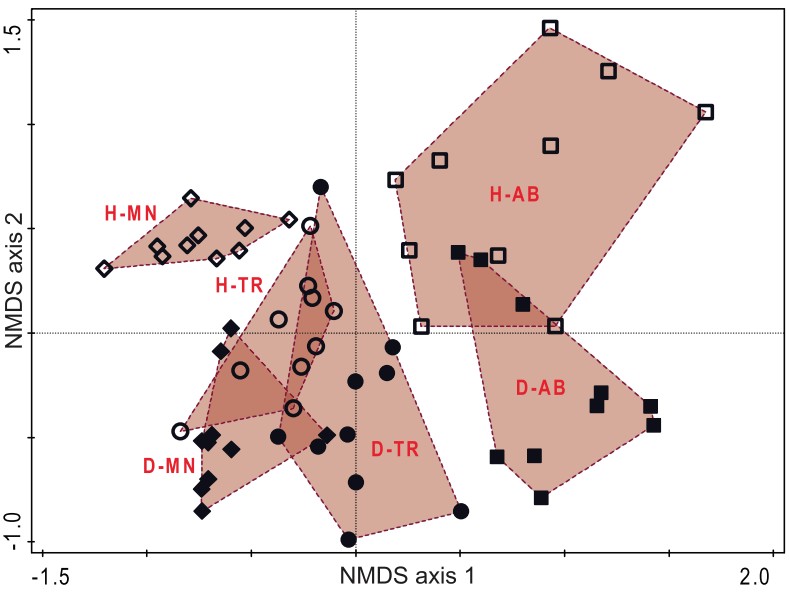

**Figure 5 Ordination diagram based on phylogenetic distance of plots.** Plot distribution according to NMDS ordination based on the UniFrac index distance matrix. Empty symbols represent hay meadows and full symbols represent dry grasslands. Rhombus represent managed plots, circles transition plots and squares abandoned ones. Convex hull envelopes enclose plots according to Habitat type and Stage of Succession. H-MN, managed hay meadows; H-TR, transition hay meadows; H-AB, abandoned hay meadows; D-MN, managed dry grasslands; D-TR, transition dry grasslands; D-AB, abandoned dry grasslands.

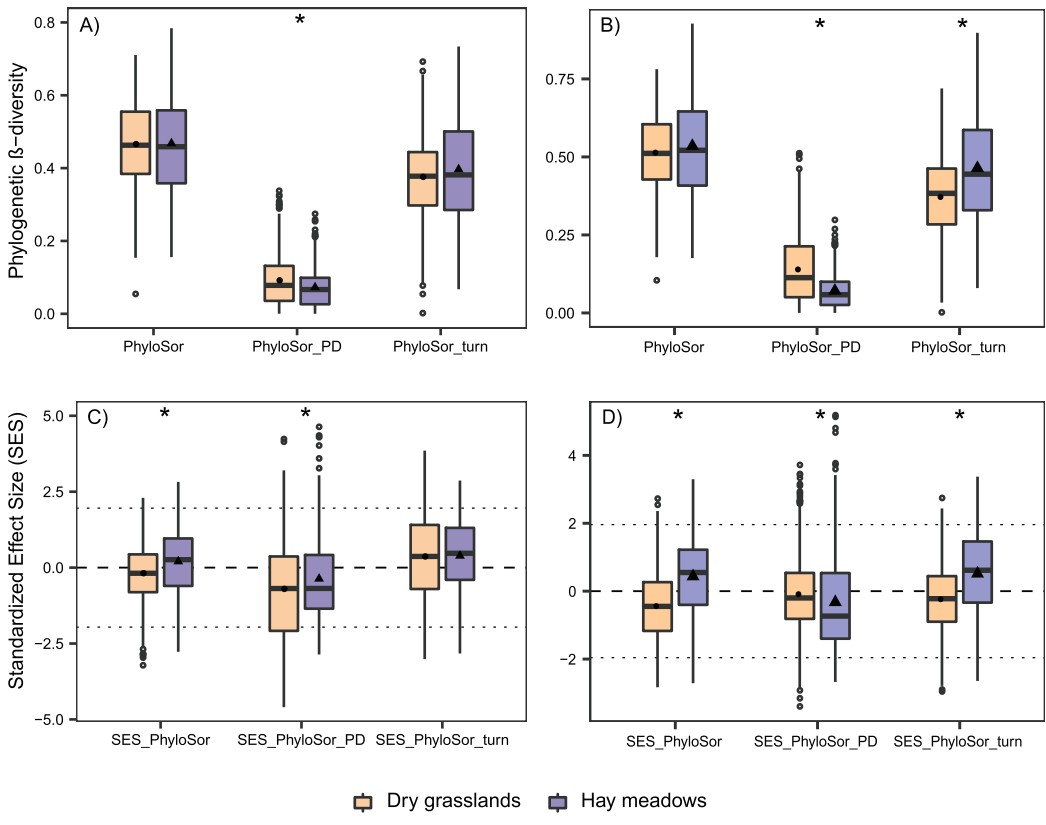

**Figure 6 Boxplot graph for the variation in phylogenetic β-diversity according to Habitat and Succession for dry grasslands and hay meadows during the succession, evaluated with and without the species *Juniperus communis*.** (A and B) PhyloSor index and its two separate components accounting for 'true' phylogenetic turnover (PhyloSorTurn) and phylogenetic diversity gradients (PhyloSorPD) evaluated with and without the species *J. communis*, respectively; (C and D) their relative Standardized Effect Size (SES.PhyloSor, SES.PhyloSorTurn, SES.PhyloSorPD) again evaluated with and without the species *J. communis*, respectively. Boxplot couples marked with * show data significantly different at P < 0.05 according to a Kruskal–Wallis test performed between the two habitats.

# DISCUSSION

The two habitats showed different features characterizing the changes in species composition during the succession, even if they displayed a comparable tendency in species richness loss. The trend in species loss has a long history of detection in the succession following the abandonment of secondary grasslands (*Corazza et al., 2016*; *Csergő, Demeter & Turkington, 2013*; *Dengler et al., 2014*; *Janišová et al., 2011*; *Rocchini et al., 2006*). Concerning Apennine hay meadows, *Ubaldi (2003)* reported that when agricultural activities are abandoned, soil water capacity and structure decreases. Consequently, also hay production decreases and these areas are used as pastures. Meanwhile, trampling and grazing further reduce water capacity and fertility of soils, so that they become dry grasslands, more or less xerophilous depending on the site. However, this general trend is not consistent with our findings, which highlighted a clear divergence of the two habitats during the succession.

The values of PD.ses showed that whether in managed and transition stages the PD is substantially consistent to what should be expected given the taxa richness, in the abandoned ones PD values were higher than expected. This indicate that whether SR decreased, PD do not decreased consistently, because the loss in species seems to be counterweighted by the presence of species with long branches in the phylogeny (and this is particularly true for dry grasslands).

Regarding phylogenetic structure, we detected for the first succession stages a predominant role of random processes of species loss in both habitats. These trends may be consistent with a framework in which a random phylogenetic structure of the community is the result of competitive exclusion of species in the case of convergent traits (see *Davies, 2006*). Nevertheless, we detected a net difference of the community structure concerning the deep nodes of phylogeny, linked to the overdispersion in the dry grasslands. This is the result of the reduction of the number of species and the appearance (as a dominant participant) of the stress-tolerant species *J. communis* (*Pierce et al., 2017*). This is in strong agreement with our hypothesis that these environments are dominated by a strong component of ecological stress. Accordingly, species assembling processes may have selected for traits allowing to survive in xeric environments (i.e., traits linked to conservative economics in the leaf economics spectrum, such as small and thick leaves, low growth rate, small specific leaf area and high leaf dry matter content), that in this case were shared between species distant in the phylogeny.

The analysis of β-diversity trends confirmed these differences. In both habitats, we evidenced a strong loss in species, but with a high component of richness difference in dry grasslands, indicating a smaller replacement by new species. These findings match with those of several authors, who showed how short species are outcompeted by dominant tall grasses in the first succession stages after the abandonment (*Corazza et al., 2016*), leading, in agreement with *Grime (2001)*, to the exclusion of subordinate and accidental species. Furthermore, in the latter succession stages many more species are lost from the community with the dominance of *J. communis*. Also in hay meadows, the loss in species following the abandonment was characterized by a suppression of species, but with a major turnover of species. Accordingly, *Csergő, Demeter & Turkington (2013)* demonstrated that the loss in species following the meadows abandonment may be driven by the suppression of dominant grasses by tall forbs, in meadows co-dominated by competitive stress-tolerant ruderals, whereas in meadows dominated by a single stress-tolerant competitor, diversity loss resulted from increased abundance and biomass of the dominant grass.

These trends were confirmed by ISA results, which are consistent with the scenario outlined in the DCA and CCA scatterplots. The higher rate of richness difference and the lower species replacement of dry grasslands, in particular emerged from the low number of species characterizing transition and abandoned stages. Indeed, in the managed stage, indicator species resulted numerically comparable among the two habitats. In managed hay meadows, the indicator species belong mainly to the families *Poaceae* and *Fabaceae* (*Lolium perenne, Phleum bertolonii, Bromus hordeaceus, Cynosurus cristatus, Trifolium pratense, Vicia sativa* and *Lathyrus pratensis*), strictly linked to the pastoral

activities, or consisted of other mesotrophic plants favored by grazing (i.e., *Ranunculus bulbosus*). On the other hand, the indicator species of managed dry grasslands were more typical of shallow and rocky soils, such as *Thymus longicaulis, Acinos alpinus, Trifolium scabrum, Bupleurum baldense* and *Cerastium brachypetalum*. In the Transition, hay meadows were characterized by a higher number of indicator species, with species typical of open habitats (*Centaurea nigrescens, Achillea collina, Poa trivialis, Dactylis glomerata, Cirsium tenoreanum*). These features are consistent with the scenario described above following *Csergő, Demeter & Turkington (2013)*. Transition dry grasslands were characterized by the dominance of *Brachypodium rupestre* (together with plants considered precursory of more closed and woody habitats, see for e.g., *Assini et al., 2014*). The dominance of *Brachypodium* spp. in successions post-abandonment is a general trend widely demonstrated for Apennine grasslands (*Corazza et al., 2016*). Finally, *J. communis* resulted the sole indicator species of the abandoned dry grasslands, while abandoned hay meadows were characterized by a high number of indicator species, spanning from various woody species of several families (i.e., *Rosaceae, Salicaceae, Aceraceae, Oleaceae*) to some herbaceous plants (e.g., *Ranunculus lanuginosus* and *Valeriana officinalis*).

The analysis of phylogenetic distances among plots highlighted that while the two habitats were well differentiated in their managed stages (in term of species lineages), a certain degree of phylogenetic similarity could be observed between transition hay meadows and both managed and transition dry grassland, due to the replacement of lineages along the succession. Nevertheless, as the succession proceeded further, the species composition of abandoned grasslands led to a net differentiation of the two habitats. Indeed, the two habitats hosted some species in common (or at least some species sharing common lineages) in the transitional stage (as also highlighted by the CCA). Subsequently they differentiated again in the last stage, with a higher replacement in hay meadows and dry grassland more or less dominated by a specie (*J. communis*) not present in hay meadows and phylogenetically distant from all other species.

As to phylogenetic β-diversity, both habitats showed a greater contribution of "true" phylogenetic turnover (reflected in a minor importance of phylogenetic gradient). These results indicated that the difference among plots was due to the replacement of species coming from different lineages, rather than from a simple difference in PD. Nevertheless, in both the cases of the simple and the standardized indices, we recorded differences concerning the amount of beta diversity deriving simply from a difference in PD. This component is numerically higher in dry grasslands (referring to simple Phylosor.PD) but is lower than what could be expected given the taxa dissimilarities when looking at the SES.Philosor$_{PD}$. We can hypothesize that this dependance may be related to the presence in dry grasslands of lower species replacement, with the entrance of a species phylogenetically very distant from the others (*J. communis*), which balances out the importance of simple PD component.

The pivotal role of *J. communis* in dry grasslands emerged on re-running all analyses concerning phylogenetic α- and β-diversity excluding this species. This species resulted responsible for a high amount of PD.ses in dry grasslands and also the main one

responsible for the presence of patterns of overdispersion. Also looking at diversity, once the balancing effect exerted by *J. communis* had been removed, the PhyloSor$_{PD}$ resulted even higher. In addition, the SES.PhiloSor component rose significantly, showing that the amount of SES.PhiloSor$_{PD}$ was substantially consistent to what could be expected given the taxa dissimilarities and is higher than the one observed in hay meadows. In addition, differences among the two habitats in both PhyloSor$_{Turn}$ and SES.PhyloSor$_{Turn}$ became significant when excluding *J. communis*, indicating firstly a lower replacement of lineages in dry grasslands, balanced by the bigger contribution of PhyloSor$_{PD}$ and secondly that replacement in dry grasslands was driven by species sharing a closer lineage than those in hay meadows. These results highlighted the important role of *J. communis* and pinpointed the importance of including this species in the analyses, also considering that this is a key species in late succession stages of dry grasslands, being the dominant one, but also that its presence may hide phylogenetic differences between habitats driven by other clades. It is also noteworthy that, whereas it is common to remove highly phylogenetic distinct species from phylogenetic analysis, this practice may lead to a partial understanding of the processes at work and that an in-depth interpretation of phylogenetic patterns should be made both using or not this distantly related species.

In this study, we used presence/absence data and it should be acknowledged that abundance data may have led to significantly divergent results. Even if in our case, one of the main species responsible for the recorded patterns was a very abundant and dominant one. Nevertheless, further studies including abundance data are necessary to better depict the processes at work.

## CONCLUSIONS

In conclusion, our data elucidate the differences in the secondary succession of dry grasslands and hay meadows in the Tuscan Apennines. In both cases, we recorded a drop in taxonomic α-diversity during the succession, but the analyses of taxonomic β-diversity highlighted quite different compositional changes, with dry grasslands mainly dominated by richness difference and hay meadows characterized by higher species replacement. As regards the phylogenetic patterns, we were able to verify that they followed a comparable trend in the earlier succession stages of the two habitats, but the entrance of a single species characterized by a deep separation in the phylogeny of the communities (i.e., *J. communis*) raises substantial differences. We propose an important role of the ecological factors in these trends, with the selection of *J. communis* fostered by a dominance of abiotic filters and resulting in the outcompeting of subordinate and accidental species in the latter stage of the habitat succession in dry grasslands, after an initial dominance of competitive exclusion of the species. Nevertheless, as noted in the introduction, the assessment of links among phylogenetic patterns and ecological processes needs more in-depth study. Our proposed scenario could be appropriate in the case of traits not conserved in the phylogeny (and this could be in accordance with the idea that trait conservativism should not be taken for granted; see *Gerhold et al., 2015*), but a certain evaluation of plant traits is necessary to further explore such trends, especially considering the concerns related to the use of phylogeny as proxies for community assembly mechanisms.

## ACKNOWLEDGEMENTS

We wish to thank the Raggruppamento Carabinieri Biodiversità - Reparto Carabinieri Biodiversità Pieve Santo Stefano, for its support. We also wish to thank the Editor Cajo ter Braak, Lars Götzenberger and one anonymous reviewer whose suggestions greatly helped to improve the manuscript.

### Funding

The authors received no funding for this work.

### Competing Interests

Renato Benesperi is an Academic Editor for PeerJ.

### Author Contributions

- Lorenzo Lazzaro analyzed the data, prepared figures and/or tables, authored or reviewed drafts of the paper, and approved the final draft.
- Lorenzo Lastrucci conceived and designed the experiments, performed the experiments, authored or reviewed drafts of the paper, and approved the final draft.
- Daniele Viciani conceived and designed the experiments, performed the experiments, authored or reviewed drafts of the paper, and approved the final draft.
- Renato Benesperi conceived and designed the experiments, performed the experiments, authored or reviewed drafts of the paper, and approved the final draft.
- Vincenzo Gonnelli performed the experiments, authored or reviewed drafts of the paper, and approved the final draft.
- Andrea Coppi analyzed the data, authored or reviewed drafts of the paper, and approved the final draft.

### DNA Deposition

The following information was supplied regarding the deposition of DNA sequences:

The new sequences are available at GenBank: MH325939; MH325941; MH325945; MH412925; MH325940; MH325948; MH325950; MH325946; MH325947; MH325942; MH325943; MH325944; MH325949.

### Data Availability

Sampled plot raw data and diversity indices, the community presence/absence matrix data, and the phylogenetic tree in Newick format is available in the Supplemental Files.

### Supplemental Information

Supplemental information for this article can be found online at http://dx.doi.org/10.7717/peerj.8683#supplemental-information.

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
