# Peer review of "Patterns of change in α and β taxonomic and phylogenetic diversity in the secondary succession of semi-natural grasslands in the Northern Apennines"

_PeerJ, doi:10.7717/peerj.8683_

## Round 0.1 · original submission · Major Revisions

As the main difference is in Juniper, I would suggest to add analysis excluding this species. The importance of this species is mentioned rather late. The CA analysis looks fine, but the 100% explained needs attention.

Reviewer 1 ·

Basic reporting

Review of Lazzaro et al. “Secondary succession of semi-natural grasslands..”

Dear authors

You study different measures of alpha and beta diversity in different successional stages within two types of grasslands. You find among others, decreasing relatedness in later successional stages, in particular in the dry grasslands. Much of this trend can be explained by the presence of Juniperus.
I appreciate that a lot of field work and data analysis went into this manuscript. But I am sorry to say - I understood extremely little in this manuscript.


First, the writing is impenetrable to me. There are likely hundreds of problems throughout the ms. Below just as an example the Abstract, it goes on like this throughout the ms. My comments in “## ##”. I am asking a lot of questions of comprehension. I can sometimes *guess* the response, but this is not my role as a reader. Note that I only comment on the writing, not on the content. Content was above. Note also that I have not left out anything from the Abstract. In many places you seem to have missed words.
* * *
“We studied the secondary succession in secondary semi-natural grasslands (dry grasslands and hay meadows) located in the Eastern side of the Tuscan Apennines (Tuscany, Central Italy). We compared these habitats, verifying: i) the patterns in species richness, composition and phylogenetic diversity during the succession; ii) whether these changes..
## which “changes”? This word does not appear before.##
.. determined ..
## how can a “change determine” something?##
..different trends in species loss and species turnover in both taxonomic and phylogenetic diversity..
##what is a “trend in species loss in a diversity”? Other than species loss.## .
…We performed a stratified random sampling, in a full factorial design along habitat type and succession stage (60 sampled plots, 10 x 2 types of habitat x 3 stages of succession). We built a phylogenetic tree of the plant communities and compared the differences in taxonomic/phylogenetic α- and β-diversity during the succession..
### compared to/between what?### .
…Both habitats displayed a decrease in species richness..
##decrease of sr with what?##
..., with a random species assembly..
##how did you study “assembly”? A diversity measure is a pattern, “assembly” is a process## ..
..in the earlier succession stages, but a shift towards phylogenetic overdispersion characterized the deeper nodes ..
## how does “shift of dispersion characterize a node”?##
..of the latter stage of dry grasslands. As to β-diversity,..
##starter only makes sense if you had before something explicitly on alpha diversity, which you didn’t.##
.. dry grasslands resulted..
##??##
..mainly dominated by richness difference..
##??##
while hay meadows displayed a higher species replacement..
##between what and what?##.
Accordingly, they..
##who, “they”?##
..showed similar..
## between what and what?##
features in phylogenetic β-diversity, with a greater contribution of the difference in number ..
##how can a “difference in number contribute anything?##
..of species sharing the same lineage in dry grassland..
##species do not share a lineage as if lineage was something acquired. Anyway, how is many species “sharing” the same lineage different from low phylogenetic diversity?”###,
..even if this trend...
## which trend?##
...is slightly lower than expected by chance...
## by chance alone you should expect a trend with advancing succession?##
due to the a very distant..
##how "distant"? Why does a distance render a trend weaker than expected by chance?##
..species (Juniperus communis).”
* * *
I understand that the above exercise might appear strange to you and the editor. I just tried to illustrate in how I literally do not succeed in penetrating this text. And it goes on like this. I fear the problem is less one of English as a foreign language (which it is also for me). For most of the sentences I commented on, I do not have the impression that any foreign-language equivalent would make sense. It is more a matter of getting your ideas 100percent straight and then using words to express them. Which is hard work for all of us. Please do not take it as an arrogance: I think you need the help of someone more experienced in this kind of work. Otherwise your hard field and stats work will be lost, which is a pity.

Experimental design

Second, why do you study these patterns, what do you learn from them?
In the Introduction or Abstract there are no biological mechanisms, let alone hypotheses. There are just descriptions of patterns, without prior proper justification of what one learns from such patterns or even predictions of patterns to be expected should a particular hypothetic mechanism operate. There is only a vague statement that we know little about some types of patterns, and that many people now say that these patterns are great to study: lines 58 ff: “Nevertheless, less is known regarding the changes occurring in the communities focusing on trends in β-diversity phylogenetic relationships, also concerning their link with changes at the taxonomic level. Indeed, according to Chase & Myers (2011) beta-diversity can provide considerable insights into the importance of deterministic and stochastic processes in generating community structure along spatial and ecological gradients. In addition, the use of molecular phylogenies may be helpful in analyzing the forces that influence patterns of biodiversity and biogeography, and in depicting the interactions among co-occurring species (Selvi et al. 2016). Indeed, in the last decades, the use of molecular phylogeny is widely increased for ecological purpose.” Etc etc.
The results present many types of analyses that are not even announced in the Introduction – so I have even less ideas what possibly these results might be good for: the CCA analysis, the SDR plots.
The Discussion then does not resolve this mystery on why studying these patterns, either. It talks a lot about results from other studies, and about the role of Juniperus. If at the end the patterns are mainly driven by the *specific* case of Juniperus then the authors should reconsider presenting a study on *general* patterns of phylogenetic diversity. I admit that this kind of criticism may be applicable to many studies on phylogenetic patterns. The Discussion does not conclude about any mechanism, either.

Third, given the two above fundamental problems I did not go into full details of the rest of the ms. Only some points pop up:
- you standardized PD by species richness, but apparently not NRI or NTD. At least NTD is strongly related to species richness.
- lot of methodological info is missing on how you standardized your measure

Validity of the findings

- You argue that “loss in species seems counterweighted..” to result in some PD.ses - but if PDses is independent of spp richness, then loss of spp is not an issue anyway

- I cannot assess whether there are some other environmental variables or spatial positions that are related to "successional stage", and that might be the true drivers of the patterns you find. You do not provide environmental information other than successional stage.

·

Basic reporting

As far as I can tell as a non-native speaker myself, the ms is well written and I didn't have any problems following the text. However, there seemed to have slipt in small grammatical errors here and there, so I would suggest having the ms proofread by a native speaker or professional service. A few examples of this:

line 34: by chance due to the a very distant species (omit "the")
line 82: these ecological difference (differences)
line 308: many more species are loss from the community (lost)

Other than that I have no additional comments, as the relevant literature is cited (though it could be extended, see comment in section 2), the ms is well structured, graphs and tables are appropriately prepared. Raw data and calculated diversity indices are provided in the form of an excel file and the phylogenetic tree of the studied species (in Newick format). I was able to open these data files, and also able to read in the Newick tree into R. The only (very minor) thing that could be mentioned here is that the names in the vegetation data are abbreviated but full names are given in the Newick tree so that it is not 100% straight forward if one would wants to reanalyze those data.

Experimental design

The study is mainly descriptional, investigating patterns of taxonomic and phylogenetic diversity (alpha and beta) along a successional gradient in two different habitats. The study therefore implicitly fills the knowledge gap of reporting and describing these pattern for the studies system, though this is not explicitly stated. In my opinion, the study could be more hypothesis-driven, or at least add some expectations in terms of the pattern of taxonomic and phylogenetic diversity patterns observed along the succession gradient. There are various hypotheses proposed regarding how taxonomic and phylogenetic diversity should change along such a gradient, and I think the authors could provide a bit more background here. The following articles and literature cited therein should provide some basic context in which the study could be set:

Letten, A. D., D. A. Keith, and M. G. Tozer. 2014. Phylogenetic and functional dissimilarity does not increase during temporal heathland succession. Proc Biol Sci 281.

Purschke, O., B. C. Schmid, M. T. Sykes, P. Poschlod, S. G. Michalski, W. Durka, I. Kühn, M. Winter, and H. C. Prentice. 2013. Contrasting changes in taxonomic, phylogenetic and functional diversity during a long-term succession: Insights into assembly processes. Journal of Ecology 101:857-866.

Verdú, M., P. J. Rey, J. M. Alcántara, G. Siles, and A. Valiente-Banuet. 2009. Phylogenetic signatures of facilitation and competition in successional communities. Journal of Ecology 97:1171-1180.


There is one aspect of the study where I am uncertain about the meaningfulness of its inclusion, and this is the part regarding the indicator species analyses. These results are nowhere mentioned in the abstract and seem somewhat tagged on to the rest of the analyses, or at least I fail to see how they are an integrative and complementary part to the other analyses, in the light of the questions put forward.

The study relies on observational data and the sampling of the vegetation is well described and appropriate for the study questions at hand.

I have to say that I do not have experience in building a phylogeny from scratch from molecular data, so I can only comment on this part that the description given in the main text as well as in the Appendix seems very detailed and well documented.

Regarding the statistical analyses, I would suggest adding some detail that is important to understand exactly how the data was treated:

- You mention mean pairwise distance as the basis for calculating NRI, but you don't specify what exactly is calculated. One could infer it from the name, but I think it would be more helpful to describe it in a simple sentence. The same goes for MNTD (mean nearest taxon distance), which is the basis of NTI. Importantly, MPD and MNTD are actually the same as NRI and NTI, but with an opposite sign. I personally think that it makes much more sense to use MPD and MNTD (or their SES) directly, as they more logically translate into over- and underdispersion. But I realize that NRI and NTI are also often used so that it is up to the authors which they prefer.

- The null model to randomize communities (for the alpha phylogenetic diversity) is merely described as "regional" null model. As this is an essential part of any null model-based study, I would prefer to read a sentence or two that describes what this null model does, instead of just referring to a published article.

Validity of the findings

The results of the study are valid within the context of the study question (but see my above point that it would be nice to contextualize the questions with some expectations/hypotheses). Especially interesting is the influence of a single species on the results and their interpretation and the authors rightfully advise to take into account such influential species and compare results with analyses excluding such species. This also provides some support for the general difficulty to infer processes from patterns.

As mentioned in the basic reporting section, the raw data are provided. Due to time constraints, I have not attempted to reproduce the results, but it would certainly be possible with the provided data (though somewhat complicated by the mentioned mismatching of species names between vegetation and phylogenetic data).

Additional comments

The authors present a well carried out study presenting a set of interesting questions. The only main objection I have is the mentioned lack of putting forward some hypotheses or expectations connected to the described diversity patterns. I think this could even add more relevance and importance to your finding that inclusion/exclusion of particular species or clades can make a big difference for phylogenetic diversity patterns, and that linking processes to patterns is often difficult and not straight forward, as also anticipated in the recent literature (e.g. Gerhold et al. 2015).

I hope that the authors find my comments helpful and worth considering.

Lars Götzenberger

References:
Gerhold, P., J. F. Cahill, M. Winter, I. V. Bartish, and A. Prinzing. 2015. Phylogenetic patterns are not proxies of community assembly mechanisms (they are far better). Functional Ecology 29:600-614.

---

## Round 0.2 · Major Revisions

General:
In the response you wrote ‘In the revision we improved the hypothesis driven approach of the paper,’ . I find it hard to believe that this work was hypothesis driven. If it is not, do not try to present it as such. The descriptive statistical analyses are of interest to anyone interested and for later meta analysis.

Apart from the major comment by the reviewer, please revise also according to the suggestions under Details.

Details:
L19 ‘verifying the patterns’ does not make sense to me if the patterns are not specified a priori.
L20 same for ‘different trends’
L26 ‘evaluating the role of the indicator species in the compositional changes’ -> We identified indicator species for each successional stage and habitat.
L28 In the analysis, differed the regional species pool of H from that of W? If the pools were identical, the CA/CCA analysis showed that species selection is nonrandom from the start.
L30 ‘resulted dominated’ : difficult reading, rephrase
L30. ‘dominated by richness difference’. Fig 3 and Fig S2 show both a big turnover in species and a richness shift.
L30 ‘ As to β-diversity, dry grasslands resulted dominated by richness difference,’-> In both habitats, the richness decreased with successional stage; in hay meadows more new species came than in dray grasslands. The latter became dominated with J. c.’ .
L33 ‘a greater contribution of the difference in number of closely related species in dry grasslands,’ I do not understand. Say simpler.

L99 peculiar? Rephrase.

L105-106. The aim stated here makes all phylogenetic analyses superfluous. Reconsider.
L118 Verify. See above.
L124. ‘predictions’ I have much difficulty seeing what you state subsequently are actual predictions. I am happy with a modest description....
L220 adopted. Defined???
L229. Indeed? Valuable? Tendencies??? Rephrase.

L 254 and L 324-329. I now understand the previous 100%. It is the efficiency as defined here. What is the reference for this efficiency/how did you get the idea to compute and report it?
L327-329 Fig. 3 and Fig. S2: Very nice results. The horizontal axis shows that there is something in common to the succession in Dry and Hay (so assemblage during succession is not random, new species come in both D and H in AB, compared to MN), apart from the differences between D and H (vertical axis). I found your wording rather strange (fig 3 does not have species). Deleting ‘for CA ordination’ and ‘for CCA ordination’ already helps. Fig. S2 has species in it.

L327-329 I recommend to put Fig3 and FigS2 as Fig3a and Fig3b, resp. The figs summarize both the alpha and beta diversity in terms of species composition (alpha: higher richness in MN than AB, beta: higher beta in MN than in AB). I would suggest to include an ordination based on a phylogenetic distance such as weighted and unweighted UNIFRAC as in customary in microbiome analysis, so as to complement Fig. 5. This would complete the comparison ‘species composition’ and phylogeny.

Fig 3. CA axis normally do not start at 0.0 (the origin), but have (0,0) as center of the plot. Could it be that a Detrended CA has be performed. In combination with the precise layout of Fig S2, I wondered whether these analyses have really been produced in R with plots using ggplot2 as stated in the MS.
Fig.4 legend seconadary?
Figure S2. How did you chose which species to plot. Refer also to table S3.
Table S3. Add to legend. See also Fig S2.
The data (now peerj-39774-All_datasets_DEF.xlsx) need to be made a supplemental file on publication according to the PeerJ open research strategy.

Wageningen, 20 December 2019
Cajo ter Braak cajo.terbraak@wur.nl

·

Basic reporting

Figure 5: Allocation of panel letters seems to be mixed up in the figure caption, i.e. a and b are showing non-standardized diversities, not a and c. The same for the standardized indices.

Experimental design

You attempted to improve the description of the null model used to calculate standardized effect sizes for the phylogenetic diversity measures. Unfortunately, your description raises more questions than it answers. Firstly, it appears a bit vague that all the mentioned community properties are held ‘approximately’ constant, without saying what that means exactly. Secondly, and more importantly, it appears that this null model is based on number of individuals. However, you described a sampling scheme that records presence/absence of species. It is not clear how that is supposed to fit together. If this null model is applicable to presence/absence data than you have to describe how so. If it is not, you used a null model that shouldn’t be used, which obviously would translate into using a more appropriate null model.

Additional minor comment regarding methods and results:

Line 224f “It describes the relationship within species focusing,…”
This is somewhat misleading, as the index describes phylogenetic relatedness, not a relationship, and it does so among species, not within.

Validity of the findings

I have no additional comments compared to my initial review for the first submission.

Additional comments

I thank the authors for their efforts to revise the submitted manuscript and take into account my suggestions and comments. While I find the overall ms improved compared to the previous version, I am still concerned about some details, especially regarding some methodological details. I detailed these concerns in the "experimental design" section of this review.

---

## Round 0.3 · Minor Revisions

Together with the reviewer, I like your revision. Please address the minor comments, so that I can accept the paper in the next round.

Additional comments:

Line 328: The numbers have too many decimals. 0 or 1 would suffice.
Fig. 5 I suggest to add labels like H-AB (as in fig 3) to Fig.5

·

Basic reporting

There are a few minor things that crept into the revised parts of the text that need some final editing:

L318: "dramatic" I would refrain from using such a word in a scientific text. The same goes for "compared to what happened" in the next line (319).

L 320: vary -> varies

L321: "all patterns toward a phylogenetic overdispersion in dry grasslands clearly vanished" also sounds convoluted. I think you could simply replace it by "phylogenetic overdispersion in dry grasslands clearly vanished".

432ff: There is something wrong with the part of the sentence. "... during the succession the replacement of lineages in the communities brought to a certain degree of phylogenetic similarity between managed hay meadows and both managed and transition dry grassland" Please rephrase.

Experimental design

I thank the authors for clarifying the issues around the use of null models. I am satisfied now with the application and description of the new null model used.

Validity of the findings

No additional comments to previous reviews.

Additional comments

I appreciate the efforts of the authors in revising their ms. At this point, I have no further comments or questions that I feel need to be addressed. I still suggest a final minor revision to smooth out some linguistic bumps (see my comments in Basic Reporting section).

---

## Round 0.4 · accepted · Accept

In the proofing stage (or earlier) please reconsider line 35 in the abstract "True phylogenetic turnover resulted the main component" I do not understand what this sentence means. For example, change "resulted" is "was", or interchange to "The main component was true phylogenetic turnover..."